# Comprehensive Energy Analysis of Vehicle-to-Grid (V2G) Integration with the Power Grid: A Systemic Approach Incorporating Integrated Resource Planning Methodology

**Marcos Frederico Bortotti** [1,*] ⬤, **Pascoal Rigolin** [2], **Miguel Edgar Morales Udaeta** [2] ⬤ and **José Aquiles Baesso Grimoni** [1] ⬤

1   PEA—Electrical Energy and Automation Engineering Department, University of Sao Paulo, Sao Paulo 05508-060, Brazil; jose.grimoni@usp.br
2   GEPEA/EPUSP—Energy Group, Department of Electrical and Automation Engineering, Polytechnic School, University of Sao Paulo, Sao Paulo 05508-060, Brazil; pascoal@gmail.com (P.R.); udaeta@pea.usp.br (M.E.M.U.)
*   Correspondence: mfbortotti@usp.br

**Abstract:** This work aims at a comprehensive assessment of the impact of vehicle-to-grid (V2G) technology on both demand and supply sides, considering integrated resource planning for sustainable energy. By using a computational tool and evaluating the complete potentials, we divide the analysis into four dimensions: environmental, social, technical, economic, and political. Each dimension is further subdivided, allowing for a detailed characterization of the impacts across these various aspects. Our approach employs a simple yet effective algebraic method using matrices to evaluate all the elements involved in the V2G system. This case study focuses on the environmental and technical–economic aspects of integrating V2G technology into a city with industrial parameters. Our findings reveal improvements and future challenges to all four dimensions, including direct and indirect reductions in $CO_2$ emissions. However, the limited availability of specific data in the social and political scopes highlight the need for further research in these areas. This study lays the groundwork for future investigations to explore the social and political implications of V2G technology, offering significant potential for future studies.

**Keywords:** V2G; vehicle to grid; sustainable energy; technical and economic impacts; social impacts; environmental impacts; political impacts; IRP; CVFP

## 1. Introduction

Through a sustainable quest for greater efficiency and reduced environmental impact in the production, storage, and transmission of electricity, ongoing efforts encompass research, pilot projects, and the exploration of novel technologies aimed at enhancing overall efficiency. Electricity generation by more sustainable means is growing exponentially with the environmental policies adopted by several countries and organizations [1].

With the growing demand for more sustainable energy generation, solar and wind, for example, face a common problem, which is seasonality or the continuity of production. With this seasonality, energy production ends up being divided into periods, so we have a great potential to generate energy for a certain period and a shortage during other periods. Because of this, technologies used to store surplus energy and make it available during lower-production times are important to enhance the most sustainable energy production. Thus, in the future, storage energy systems will be part of the characterization of electric energy, in addition to already known generation, transmission, and distribution systems.

Different types of electrical energy storage systems (ESS) have been under study in recent years. Presently, some are more feasible than others when solely considering

technical and cost analyses. The presumed benefit of exclusively applying technical–economic analysis to establish the viability of one ESS over another is the simplification of the decision-making process. Nevertheless, there are multiple drawbacks to neglecting the examination of their social and political origins, which may only become apparent in the long run.

This paper seeks to improve the method of choosing ESS for a specific region, including the technical–economic factor, as well as environmental, social, and political dimensions. This method, considering these four dimensions, comes from the analysis of complete costs initially applied to the integrated planning of energy resources (IRP), which was improved to aid decision makers in choosing the best energy resources to be used primarily in a specific region. Due to the nature of the analysis of energy resources and energy storage being somewhat different, some attributes and subattributes contained in the dimensions needed to be adapted (some created and others excluded), bringing a greater adherence to this new application.

This methodology attempts to cover all the elements that are essential to characterize the means of energy storage with the attributes and subattributes of the four main dimensions. These attributes are distributed within the dimensions as branches according to the level of importance that they have in the final characterization (first level: attributes; second level: subattributes; third level: sub-subattributes, etc.).

Specifically, for this study, a mixed ESS method is used, given the V2G nomenclature. With the increase the use of electric vehicles (EVs), concerns with respect to increased energy consumption have become worrying. Therefore, studies on how to mitigate these impacts may be very important.

V2G offers a solution for the storage of surplus energy. With growing concerns about sustainability, solutions to environmental impacts have been of great importance. The use of hybrid and electric cars has been increasing in the world market due to this sustainable movement. Some countries and regions have incentives and public policies for the insertion of electric cars, including China, Europe, and the USA [2]. An electric vehicle has a battery as its energy source, which is charged through the electrical network. To use this asset for energy storage and transportation, the V2G aims to use the battery not only for locomotion in the vehicle but also as a generator, discharging accumulated energy back into the network [3,4].

V2G aims to enhance seasonal generation methods with the use of batteries in cars. Along with the impact that the increase in the use of electric vehicles will have on the grid, methods of improving the management of the growing energy demand are important for the evolution of the electrical system. Herein, we analyze the impacts of technology on both the demand and supply sides of the electrical system using integrated resource planning (IRP).

IRP differs from traditional planning in terms of the class and scope of the resources considered, the current participation of the owners and non-owners of the resources, the bodies involved in the resource plan, and the criteria for the selection of alternatives [5].

Our endeavor is founded on conducting an exercise involving the calculation and assessment of complete potentials, a fundamental stride in the IRP (integrated resource planning) process. This calculation and assessment draw upon preliminary data gathered through surveys and a comprehensive environmental inventory. Additionally, the identification of resources and stakeholders, as well as their interests, stands as a pivotal requirement. When the information is non-numeric, it must be converted into absolute values. Indicators are also used in the analysis, always within the four dimensions: environmental, social, political, and techno-economic [6].

For V2G technology, the supply side includes seasonal energy generation, such as small- and medium-size solar and wind generation, storing surplus energy to return to the grid when the production of these resources is low. On the demand side, the following are considered: the transport of electricity, backup for the lack of electricity, the management of peak hours, and the decrease in consumption during periods of high energy tariffs [7].

As per Leduchowicz-Municio, 2022 [8], a thorough assessment was conducted regarding the environmental and social implications of incorporating Battery Energy Storage Systems (BESS) into Medium Voltage (MV) commercial facilities. This integration aims to facilitate energy time-shifting during peak hours and function as a backup energy source during grid outages. The study aims to validate the sustainability of employing BESS-based peak power plants in MV commercial units, particularly in developing countries. The combined Resource Complete Potential Assessment-Life Cycle Inventory (RCPA-LCI) method is used, with RCPA highlighting the essential criteria for evaluating socio-environmental impacts and LCI for providing a comparative analysis of energy consumption, generation, and losses between peak plant configurations. The assessment focuses on $CO_2$ emissions reduction and job generation, addressing their significance to climate change and to social issues.

The findings reveal the potential for a substantial reduction of 15.4 million tons of $CO_2$ emissions, leading to an estimated economic saving of USD 154 million and the creation of 113 new jobs annually. The socio-environmental assessment demonstrates the positive impact of using BESS-based peak power plants, replacing fuel-based alternatives in the commercial sector. Given the established technical–economic viability, this study confirms the sustainability of BESS-based solutions.

With this, the work aims to study the impacts caused by the insertion of this technology both for the demand and supply side, computing the valuation of potentials and mapping it environmentally. As a reference for the study, the city of Santos in the state of São Paulo, Brazil, was chosen. The supposed replacement of the entire fleet of combustion passenger cars with electric vehicles with battery storage served as a database, especially regarding some sub-attributes.

## 2. The V2G Technology with Energy Resource

With the increase in the electric vehicles fleet in some countries, the utilities and agencies responsible for the operation of the electric system are faced with the increasing consumption of electricity, which led to the need to expand its production. According to the report "Studies on the 10-year Energy Expansion Plan 2030" by the Energy Research Company (EPE—Portuguese acronym), until 2030, the difference between the energy load results for the upper and lower scenarios reaches a 16 GW average, or about 29 hydraulic GW (the equivalent of two Itaipu power plants, including the Brazilian and Paraguayan capacities). This has been generating a great deal of stress on the management of the national interconnected system, serving as support for incentives in storage technologies [8].

For example, considering that the users charge their electric vehicles precisely during peak hours of energy consumption—between 17:00 and 19:00—it means the demand for energy at this time will increase. As the demand for electricity at peak times is already a problem for the electrical system, that increase can derive complications. This leads to the need for reinforcements in the system to prevent overloading and even an increase in the tariffs due to the higher production cost. This unexpected consumption can be problematic for the electrical system operation and thus lead to failures in the supply of electrical energy, resulting in an acceleration of the generation expansion process [9].

There is a demand to create a technology to smooth spikes in consumption (peak shaving) and store the electricity generated by alternative sources, besides maintaining a more balanced energy consumption curve throughout the day. This is when the V2G technology emerges, as shown in Figure 1. This technology is based on turning an electric vehicle into a source of energy that would plug into the grid to take advantage of the battery present in these electric vehicles to store and thus discharge the electricity into the electrical grid when necessary.

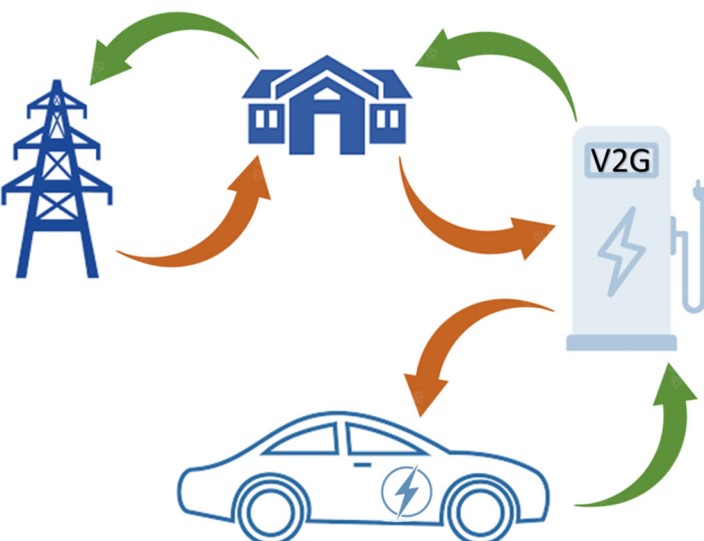

**Figure 1.** Vehicle-to-Grid technology (Source: the authors). Orange represents the electricity flowing from the grid to the EV, while green symbolizes the electricity flowing from the EV back to the grid.

For operationalizing this conceptual framework, an array of daily investigations were undertaken, encompassing diverse tariff stratifications contingent upon temporal utilization patterns. These distinct tariff tiers are intricately linked to discrete diurnal intervals and their concomitant power origination modalities, including those harnessed through the deployment of photovoltaic arrays seamlessly integrated within residential consumer environs.

Additionally, the integration of Vehicle-to-Grid (V2G) technology has been envisaged to facilitate bidirectional energy exchange. This entails the injection of surplus vehicular-derived energy into the overarching distribution grid during periods of elevated demand, juxtaposed with the judicious consumption of grid-derived energy during epochs of diminished electrical requisition. This strategic energy management paradigm correlates with temporal fluctuations in energy pricing, optimizing expenditure through consumption during troughs in demand and the corresponding abatement in energy prices.

A predominant thrust of ongoing scholarly inquiries is fixated upon the evaluative metrics of this technological stratagem, which encompasses not only performance evaluations, but also the discernible mitigation of greenhouse gas emissions. Moreover, the assessment pertains to the transformative impact on extant electrical infrastructure, with pronounced emphasis on network ameliorations. Concurrently, analytical focus extends to the consequences of intelligent charging paradigms, typified by the V2G framework, whereby perceptible enhancements in systemic operations are manifest (refer to Table 1).

Note that, amidst these salient advancements, an urgent research lacuna persists regarding the broader ramifications in societal and ecological structures. Despite the advances achieved thus far, a distinct imperative subsists and needs comprehensive exploration of the ramifications of this technology in existing socio-environmental frameworks.

As for the vehicular exemplar chosen, the midsize electric car predominates within this discourse. The rationale for this selection is grounded in the pervasive market presence of the midsize electric car in jurisdictions that have embraced this technological trajectory. Noteworthy attributes of the chosen vehicle encompass a battery capacity of 40 kilowatt-hours, which concomitantly endows an estimated travel range of 389 km as per the New European Driving Cycle (NEDC) urban assessment. Moreover, the vehicle charging dynamics are underscored, signifying a maximum charging duration of 8 h, as elucidated in Table 2.

**Table 1.** Studies on the impacts of smart charging technologies.

| Study | Scenario | Uncontrolled Charging | Smart Charging |
|---|---|---|---|
| **IRENA, 2019 [10]** | 50% penetration in an isolated system with a 27% solar share | 9% increase in peak load and 0.5% solar curtailment | 5% increase in peak load (V2G) and down to 0% curtailment |
| **RMI, 2016 [11]** | 23% penetration US (California, Hawaii, Minnesota, New York, Texas) | 11% increase in Peak Load | 1.3% increase in peak load (V2G) |
| **Taljegard, 2017 [12]** | 100% penetration in Denmark, Germany, Norway and Sweden | 20% increase in peak load | 7% decrease in peak load (V2G) |
| **McKenzie, 2016 [13]** | 50% Penetration in the Island of Oahu, Hawaii, US 23% VRE share | $10 \times 23\%$ VRE curtailment without Evs | 8–13% VRE curtailment with smart charging Evs |
| **Chen and Wu, 2018 [14]** | 1 Million Evs in the Guanzhou region, China | 15% increase in peak load | 43–50% reduction in valley/peak difference |

Source: the authors, assembly based on [10].

**Table 2.** Electric vehicle recharge times.

| Recharge Times | | |
|---|---|---|
| Emergency Cable | Wall Box | Fast Charge |
| 6.6 kW | 6.6 kW | 50 kW |
| 40 kW 100% | 40 kW 100% | up to 80% |
| up to 20 h | up to 8 h | up to 40 min |
| Standard Connection | Standard Connection | Fast Charge Connection |

Source: The authors, assembly based on [10].

## 3. Full Characterization of the V2G

The main purpose of this paper is to use CVFP (Computation and Valuation of Full Potentials) as a tool for valuing V2G into attributes, specifically for the city of Santos, state of Sao Paulo, Brazil.

The other objectives are:

-   Suitability of the attributes and sub-attributes, derived from the generation of energy, for use with ESS;
-   Characterization of the impacts at the environmental, economic, social, and political levels of this technology;
-   Viability of the technology for users and energy distributors, as well as opening a range of discussions of possible alternatives to the current mobility system.

For classifying different items, it is initially necessary to know them. Therefore, all the elements under analysis must be characterized. The method in question seeks to characterize the different ways of storing energy having the four main dimensions as a base. Each of these dimensions has attributes and sub-attributes at different levels, according to their degree of importance. This means that each dimension can be represented by a priority diagram.

### 3.1. Techno-Economic Dimension

The techno-economic dimension is composed of attributes that characterize the types of electric energy storage considering technical and financial factors.

It is formed by three main attributes: Reliability, Technical Facility and Costs, which are divided into sub-attributes (refer to Figure 2).

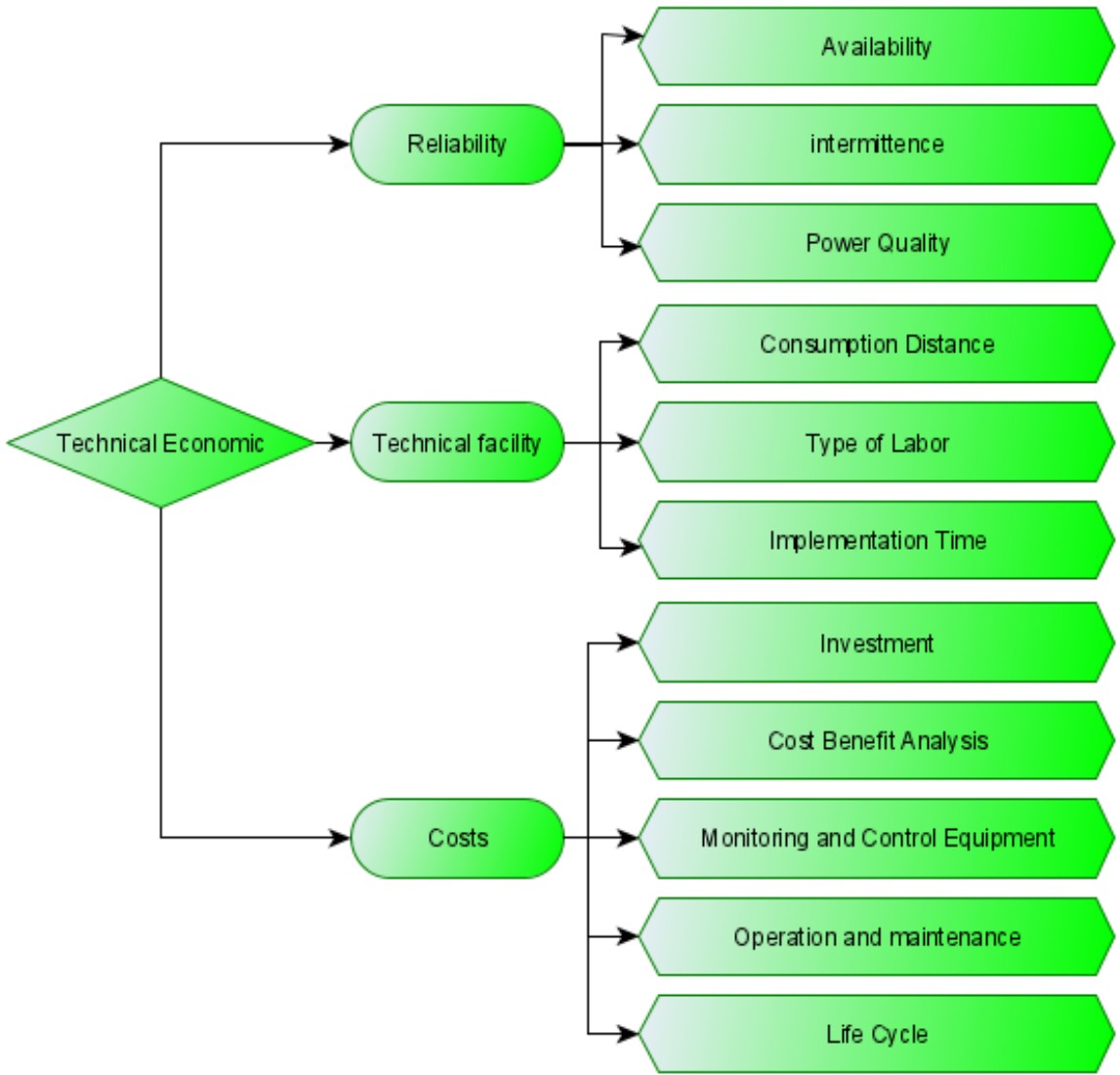

**Figure 2.** Tree of technical–economic dimensions (Source: the authors).

A.　Reliability

The "Reliability" attribute is defined by the system's ability to perform and maintain its operation in routine circumstances or in hostile and unexpected circumstances. This attribute is divided into three sub-attributes, as follows:

- Availability: The percentage of time the equipment is available to perform its function, excluding the time when stopped for maintenance or any other reason.
- Intermittency: This attribute is related to the technology availability time for the network.
- Power Quality: It is measured by the equipment capacity to supply energy to the electric grid in the cleanest way possible; this implies the possibility of the system to generate undesirable harmonics or not and in what proportion.

B.　Technical Facility

The "Technical Facility" attribute is defined by the complexity of construction, operation, and maintenance of the ESS, as well as their location. This attribute is composed of three sub-attributes:

- The distance of consumption: It is defined by the location of the energy storage systems concerning their consumption center (it can be determined by the distance from the electric network it is serving). It is measured in kilometers.

- Type of labor: This attribute describes the necessary specialization of the labor that will deploy, operate, and maintain the energy storage system.
- Deployment time: This is defined by the number of months required to implement the storage system in question.

C.    Costs

The "Costs" attribute covers all monetizable costs. It is defined by the most important economic and financial tools for the case. This attribute is divided into five sub-attributes:

- Investments/Loading costs: These are measured by the amount of money needed for installing each ESS and the money needed to load each type of ESS.
- Cost–Benefit Analysis: It is a dimensionless value given by the total cost of the system divided by the benefit brought by the system considering its entire useful life in operation.
- Monitoring and control equipment: Some types of ESS require additional equipment for their proper functioning; therefore, this sub-attribute is defined by the cost of this extra equipment.
- Operation and Maintenance: Costs to operate and keep the ESS up and running in one year. This attribute is measured by the sum of a fixed cost and a variable cost.
- Useful life: Defined by the number of years that the ESS can operate for the purpose intended.

### 3.2. Environmental Dimension

In this dimension are attributes and sub-attributes that seek to characterize the ESS considering the environmental impacts caused in the three main environments: aquatic, aerial and terrestrial (refer to Figure 3).

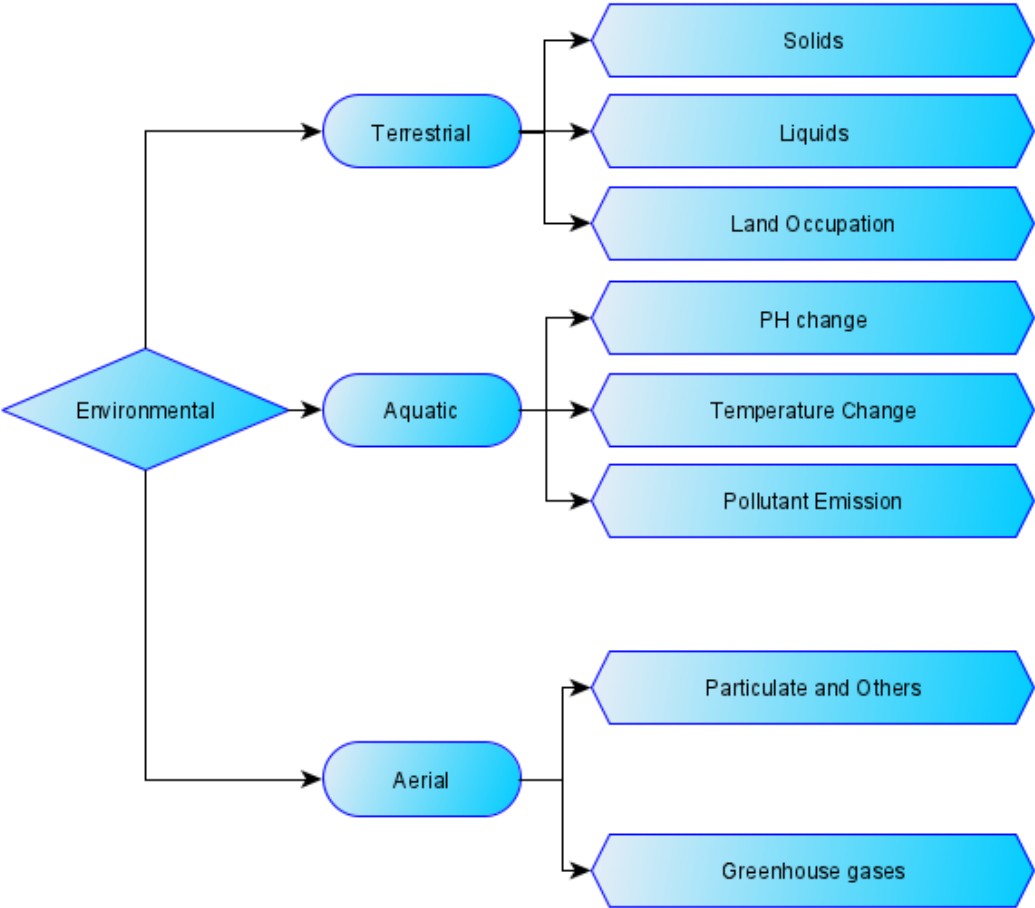

**Figure 3.** Tree of environmental dimensions (Source: the authors).

A.   Terrestrial

This attribute is divided into three sub-attributes:

- Solids: Referring to the amount of solid waste (data in kg) produced during the energy storage activity.
- Liquids: Referring to the amount of liquid waste (in liters) produced during the energy storage activity.
- Land Occupation: Area used by the given type of energy storage being analyzed (in $m^2$).

B.   Aquatic

The aquatic environment attribute is divided into three sub-attributes:

- Change in pH.
- Change in water temperature.
- Pollutant emission.

C.   Aerial

The aerial environment attribute is divided into two categories (sub-attributes):

- Particulate and others: Referring to the amount of particulate matter emission caused by the energy storage activity.
- Greenhouse gases: The amount of $CO_2$ equivalent to greenhouse gases emitted during the energy storage activity.

### 3.3. Social Dimension

The Social Dimension seeks to expose all factors that somehow change the way of life of people affected by ESS. This dimension is divided into four attributes, each with its sub-attributes (refer to Figure 4):

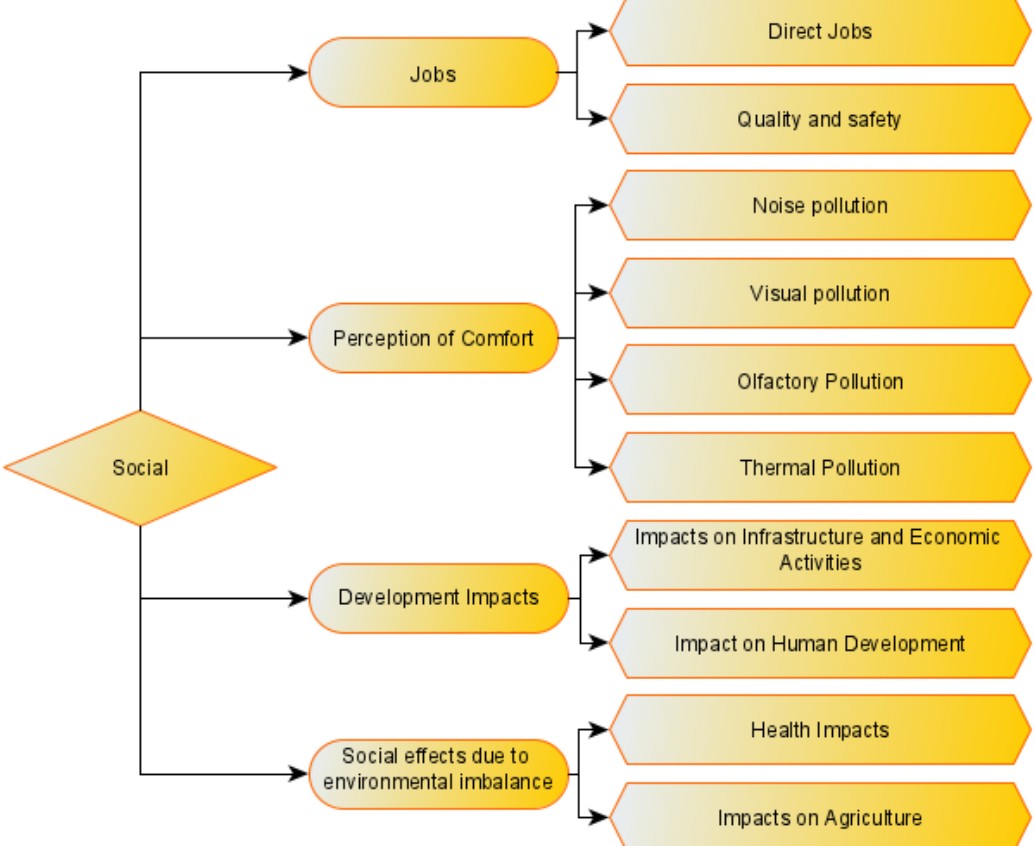

**Figure 4.** Tree of social dimensions (Source: the authors).

A.　Jobs

The Jobs attribute is divided into two sub-attributes:

- Direct jobs: The number of direct jobs generated during the construction, operation, and maintenance of the V2G.
- Quality and safety: This sub-attribute characterizes V2G based on the type of labor needed for its operation. It assesses the level of specialization required and the overall safety of the job in terms of its viability.

B.　Perception of Comfort

The Perception of Comfort attribute is defined by four sub-attributes all related to people's "feelings" concerning V2G. These are:

- Noise pollution: This attribute can be defined by the amount of dB emitted by the V2G at a comfortable distance (this varies according to the system location).
- Visual pollution: This sub-attribute is quite subjective, as it depends on the perception of each person; it can be determined through local interviews asking what people think about the implementation of a system that visually modifies the environment they live in.
- Olfactory pollution: This sub-attribute is quite subjective, as it depends on the perception of each person; it can be determined through local interviews asking what people think about the implementation of a system that olfactorily modifies the environment they live in.
- Thermal pollution: This sub-attribute is defined by the emission of heat by the energy storage system and whether this heat is sufficient to alter the environment around it.

C.　Impacts on the Development

The Impacts on the Development attribute refers to factors that can both positively and negatively impact social development activities in the region. This attribute has two sub-attributes:

- Impacts on infrastructure and economic activities: This item measures the impacts caused by the creation of new infrastructures and the development of new economic activities (creation of new businesses) due to the implementation of V2G.
- Human Development Impacts: This value is known as the regional HDI, an item that can be evaluated considering the improvements or negative effects related to V2G.

D.　Social effects due to environmental imbalance

These fall into the following sub-attributes:

- Impacts on health—considering the elimination of automobiles, the combustion within the studied city would experience health improvements due to the reduction of greenhouse gas emissions.
- Impacts on agriculture.

*3.4. Political Dimension*

This dimension takes into account several factors of guidance and evaluation, such as future needs, and the development and introduction of a particular ESS. In the political dimension we also seek to encompass different expectations about the implementation of each resource by the En-In throughout the chain and to ensure a satisfactory result for all parties involved. The political dimension is composed of three main attributes, each with sub-attributes that define them (refer to Figure 5).

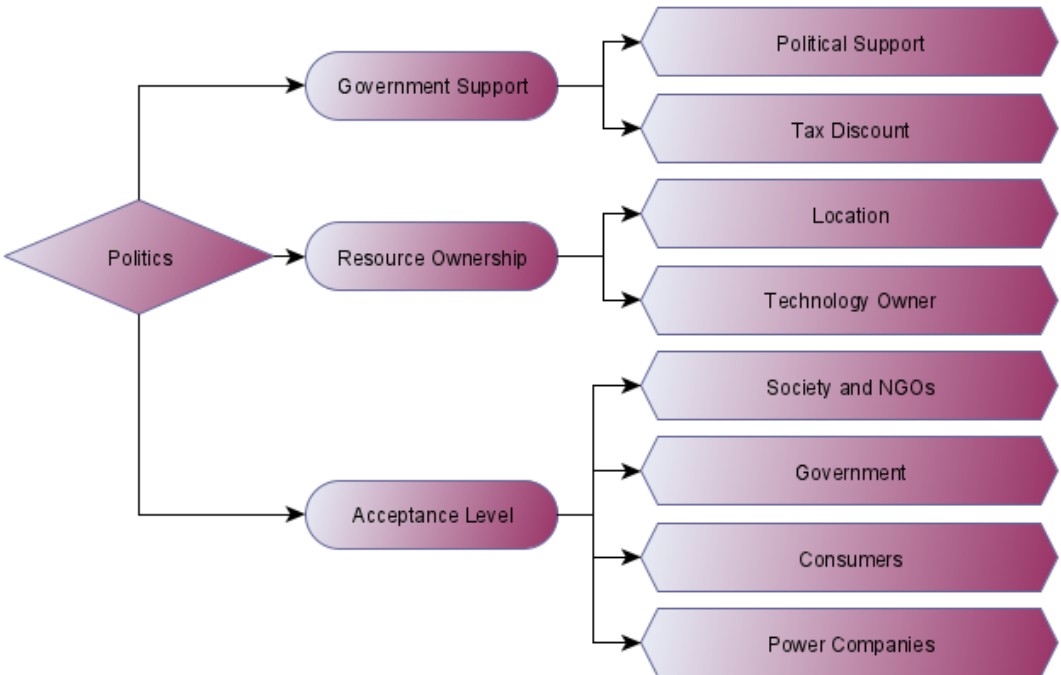

**Figure 5.** Tree of political dimensions (Source: the authors).

A.    Government Support

The Government Support attribute is composed of two sub-attributes:

- Political support: This defines whether there are incentive measures by the local government for certain ESS amounts (they can be measured by the amount of incentive or prohibition laws).
- Tax discount: The analysis aims to determine whether there are any tax incentives in the study region to promote the utilization of certain energy storage systems (ESS).

B.    Resource Ownership

The second attribute is Resource Ownership; its sub-attributes define to whom the equipment technology belongs. This can be especially important if currency conversion is required. This attribute comprises two sub-attributes:

- Location: Different from the sub-attribute "distance from consumption", this item covers more political issues. It is very important to know where the V2G technology will be implemented, whether international agreements are required to receive the stored energy, or even provincial negotiations considering that the implementation site is not the same as the location that requires the energy.
- Technology owner: this item measures the percentage of people that own the technology used in the ESS and in what proportions.

C.    Acceptance Level

The Acceptance Level is divided into the following sub-attributes:

- Society and NGOs;
- Government;
- Consumers;
- Power companies.

## 4. Full Dimensioning of V2G Energy Resource

With the dimensioning characterized, a survey of works, research, books and pilot projects was conducted to scale the CVFP for the V2G. With these data, it was possible to

characterize each dimension and its attributes and sub-attributes. The dimensions related to the implementation and use of the V2G technology are as follows.

*4.1. Technical-Economic Dimension*

A.    Reliability

- Availability: The use of technology will be more comprehensive within a household. However, we have some applications for using this technology, the main ones being frequency regulation, cutting peak consumption and mitigating voltage fluctuations caused by the high penetration of renewable energies.

    The technology will be used for cutting peak consumption, normally in Brazil from 18:00 until 21:00, totaling 3 h a day. In this use, it must allocate 365 days a year, totaling 8760 h. If used 3 h a day, every day of the year, we would have a penetration of 12.5% of availability in the year.

- Intermittency: The battery is technically 100% of the time available while it is charged. By definition, V2G is a technology used to discharge the battery energy into the electrical network. For its best use and feasibility, it is used intelligently, discharging at the most expensive energy time and charging at the cheapest time. Each time an electric vehicle (EV) is connected to the electrical grid, the primary intention of the user is to initiate battery charging. In the context of Vehicle-to-Grid (V2G) technology, the system can autonomously assess the grid energy demand and pricing dynamics, optimizing the usage of energy resources. The system performs a self-regulating analysis to determine whether the energy stored in the EV's batteries exceeds the established safety threshold, rendering it available for grid integration. If the energy level surpasses the safety threshold and coincides with peak hours, the system will proceed to discharge the surplus energy into the grid. Conversely, during off-peak hours when energy costs are lower, the system will retain the energy in a stationary state until an optimal time for charging the battery occurs [15].
- Power Quality: Several V2G factors positively influence the power quality of the network. For example, active energy regulation and stabilization, voltage regulation, frequency regulation (down–up), spin reserves (reserves that are online but out of the load and act as a support for the system in the event of supply failures—spinning reserve), reactive energy support, cut in peak consumption, filling of consumption vouchers, load following (energy backup that adjusts power delivery to the grid as needed), energy balance and harmonic current filtering [16].

B.    Technical facility

- Distance from consumption: The consumer who is connected to the distribution network supplies electricity to the electricity grid; thus, this energy does not need to be transported from a generator until it reaches the final consumer.
- Type of labor: It can be separated into three types: unskilled labor, technical knowledge with a certificate of need for higher education with a specialization in the area in question. Specialized labor is required to install a connection point to the power grid. After that, the user himself can use the plug-and-play system.
- Implementation time: Time exists, but it is minimal, not considered as a negative impact. Due to this technology, it is ready-made equipment "off the shelf".

C.    Costs

- Investments/Charging costs: The cost of installing the V2G is approximately 4% of the vehicle value [17]. For a 40 kWh battery, the cost is USD 30 per kW considering the price of the midsize electric car at USD 30,000. If we consider the price of 0.66 reais per KWh, there is a high cost per KW for the consumer.
- Cost–Benefit Analysis: It is possible to have a financial return of approximately 7% per vehicle per year (refer to Table 3) [18].

**Table 3.** Ramp down best combination results—comparison of costs.

| Parameter | | Decrease—V2G | | ICE |
|---|---|---|---|---|
| | | EV | PHEV | |
| Battery Proportion (kW) | | 99 | 10 | n/a |
| Charger Proportion (kW) | | 19.2 | 19.2 | n/a |
| SOC (%) | | 30 | 30 | n/a |
| Regulation Cycle | | 20:00–8:00 | 20:00–8:00 | n/a |
| 10 years | Price | USD 32,529,037 | USD 32,723,614 | USD 33,198,377 |
| | V2G Income | USD 2,268,780 | USD 1,758,834 | - |
| | Final Expense | USD 30,260,257 | USD 30,964,780 | USD 33,198,377 |
| Per Mile | Price | USD 0.735 | USD 0.739 | USD 0.750 |
| | V2G Income | USD 0.051 | USD 0.04 | USD - |
| | Final Expense | USD 0.683 | USD 0.699 | USD 0.750 |
| V2G income per vehicle/year | | USD 907.51 | USD 703.53 | USD - |
| Reduction on fee from V2G | | 7.0% | 5.4% | 0.0% |
| Savings vs. ICE | | 8.9% | 6.7% | n/a |

Source: the authors, assembly based on [18]. n/a—Not applied.

Considering a hypothetical midsize electric car, this vehicle with its battery system can benefit from gains with frequency regulation. For these, one can save USD $2140 (although a portion of that revenue is likely to go to an aggregator). Adding a charger with V2G capability where the car is parked during the user's/owner's working time increases annual revenue by about USD $600,00 [19].

Considering these values for the USA, it would take just 10.5 years for a homeowner to recover that value, thanks to the revenues generated by the V2G (disregarding other benefits of ownership of an EV, such as reduced fuel costs). At the same time, note that frequency regulation prices are highly variable over time and for electricity networks.

- Monitoring and control equipment: As the technology (V2G charging unit) already has all the monitoring and control equipment built-in, the cost, in this case, will be zero.
- Operation and Maintenance: This is around 5% of the cost of implementation [20].
- Lifetime: There are no conclusive studies regarding the impact of V2G on battery life. A question that remains is whether participation in the V2G also affects the battery life. In [21], it is argued that the use of vehicle batteries for V2G energy incurs approximately half the loss of capacity compared to the fast cycle found while driving. The percentage of loss of capacity (by normalized Wh or processed Ah) is quite low: 0.006% for conduction support and 0.0027% for V2G support. The analysis shows that several thousand driving days/V2G incur substantially less than 10% loss of capacity, regardless of the amount of V2G support used. The extent of the impacts can be less, zero, or even improve the health of the battery [20,22]. Disregarding the battery degradation due to the V2G and only taking into account the useful life of this battery due to the use of the car, the value is about 8 years (or according to the recharge cycle indicated by the manufacturer).

### 4.2. Environmental Dimension

As V2G is a technology that will occupy an existing space (within the installed establishment) and be just a technology with a share in the flow of energy (loading and unloading electrical energy from the grid to the vehicle), some of these environmental impacts will not take place. Considering the impact of the availability of energy to be added to the network, one can have some impacts upstream (generating), thus taking into account studies that indicate some related data [23–25].

A.    Terrestrial

- Solids: During the V2G operation, there is no production of solid pollutants.
- Liquids: During the operation of the V2G, there is no production of liquid pollutants.
- Occupied space: During V2G operation, the space occupied is minimal, and therefore not considered.

B.    Aquatic

- Change in pH. During the operation of the V2G, there is no production of pollutants to alter the pH.
- Change in water temperature. During the operation of the V2G, there is no production of pollutants to alter the water temperature.
- Emission of pollutants. For this factor, we do not consider any impact from the technology.

C.    Aerial

- Particulate and others: During the operation of the V2G, there is no production of particulate pollutants.
- Greenhouse gases: Reduction of 0.08 $MtCO_2$/year according to the Brazilian electrical matrix. Taking into account the Santos immersion, it has a proportion of 0.15% of the Santos emissions. Yet the major part would be the reduction in the fleet of cars combustion by electric cars. This is not considered because the electric cars themselves are not being considered, but rather the V2G intelligent charging technology.

*4.3. Social Dimension*

A.    Jobs

- Direct jobs: As it is an unconsolidated technology, there are still no figures corresponding to the generation of jobs in the manufacture of V2G modules. Considering the operation part of the installed system, the customers (consumers) or the energy concessionaire themselves can be considered to be in charge of this configuration/operation of the modules. High demand for skilled labor is expected for maintenance [26].
- Quality and safety: V2G modules are produced such that they have a friendly interface for the customer to handle this device.

B.    Perception of Comfort

- Noise pollution: No noise pollution is caused.
- Visual pollution: No visual pollution is caused.
- Olfactory pollution: No olfactory pollution is caused.
- Thermal pollution: No thermal pollution is caused.

C.    Impacts on Development

- Impacts on infrastructure and economic activities: Need to install energy meters with technology for distributed generation. Businesses related to the sale of technology will be created. It opens several possibilities related to the sale of energy.
- Impacts on Human Development: It can indirectly impact knowledge concerning electricity, economy and environment. This is due to the users' greater awareness of how much they will spend and how much energy they can sell to the grid. There is no relevant work on this topic about the V2G technology.

D.    Social effects due to environmental imbalance

- Impacts on health: Considering the removal of automobiles, the combustion within the studied city would contribute to enhanced health due to a reduction in greenhouse gas emissions. The impact solely attributed to V2G technology becomes less significant with the transition from combustion vehicles to electric ones [26].
- Impacts on agriculture: This type of technology does not directly affect agriculture because it takes place in urban centers.

*4.4. Political Dimension*

A.    Government support

- Political support: There are currently some incentives in Brazil concerning the implementation of EVs; these always have an appeal focused on the environmental issue. Some current incentives in Brazil are exemption from import taxes, discounts on taxes such as IPVA and IPI and exemption in Rodizio (São Paulo). As the technology studied, V2G is still under study and implementation of pilot plans; implementation support is not yet active. Since V2G has a greater integrative capacity between renewable sources and the use of distributed generation, it can be expected to attract political appeal. However, no data are yet available to prove such possibilities.
- Tax discount: There is currently no dimension of the possibilities that can be brought about with the use of V2G and how incentives can be generated. As the technology allows for a "sale" of surplus energy to the grid and makes it possible to improve energy quality, it may be the target of incentives such as tax rebates in the future.

B.    Ownership of the Resource

- Location: Not affected by the negotiation between the distributor and the consumer.
- Technology owner: A mix of national and imported equipment (in which proportion), fully national equipment and equipment fully produced in the region of demand can be imported. It is currently 100% imported, but it can generate demand for national technology as the implementation grows. Regarding technology, we have the equipment to be produced for residential installation. Conversely, the installation service will be national, with a more specialized national workforce.

C.    Level of Acceptance

- Society and NGOs: Improvement for society due to the reduction in GHGs and improved energy quality. However, the higher cost of implementation is not accessible to all levels of income.
- Governmental: These players do not have a direct relationship with technology, but they are important for its entry and growth in the market, since they can create regulations and invest in infrastructure to make the technology viable. These policy makers can help to stimulate transmission and distribution operators to create their energy storage policies and regulations. They can also make space for ESS regulation, thus supporting for the creation of tax regulations on participants in the electricity production and consumption network.

General tax settings can impact the implementation of V2G systems. Government policies can also indirectly influence the increase in renewable sources, as well as decarbonization and carbon taxes. These, in turn, have great appeal for storage technology, as seen above, improving the productivity of renewable sources and increasing their yield [19].

- Consumers: Society can improve due to the reduction of GHGs and improvement in the quality of energy, decrease in the electricity bill of demand and energy storage itself in times of lack of supply.
- Power companies: Due to the improved power quality, improvement in the DEC/FEC indexes becomes a benefit for energy companies, such as utilities. In terms of energy generators, they are mainly renewable energy generators that have a direct connection with the use of V2G technology to improve their productivity.

This technology brings greater complexity to the system, requiring large installations related to the management and monitoring of the network. Cybersecurity concerns will also be intensified, thus generating greater demand for investments [19].

The following is a summary of the results from the different dimensions. In this summary, the most important dimensions and aspects for both the demand and supply sides contemplated by the research are separated (refer to Table 4).

**Table 4.** Summary of results.

| Supply | Supply Side | Demand Side |
|---|---|---|
| Economic Technical | − Frequency regulation<br>− Peak shaving<br>− Seasonal energy storage | − Simple Operation<br>− Comfort<br>− Power quality<br>− Power backup |
| Environmental | − Possible increase in battery disposal | − Air quality improvements |
| Social | − Connectivity to the current electrical network<br>− Cybersecurity | − High cost<br>− Understanding of technology |
| Policy | − Regulation | − Investments<br>− Job creation<br>− Development of the HDI |

## 5. Conclusions

The electricity storage methods have presented great feasibility concerning seasonal power generation. Therefore, V2G technology, which has the storage of electricity in batteries, has numerous benefits to society, such as improving electrical quality, reducing environmental impacts, and providing support in case of occasional interruptions in the electricity supply. This technology contributes positively to the insertion of EVs, replacing the increase in energy consumption, which in turn increases the expenditure on electricity and thus brings the feeling of savings in the search for energy sustainability.

However, there are still numerous unanswered questions related to all the variables involving the implementation and viability of this technology. As noted, barriers still exist to the future of EV and energy storage technologies. For V2G, technical, economic, social, and regulatory barriers can be identified as points of improvement for the future. Among these, battery degradation and charging efficiency can be characterized as technical issues. On the economic side, it is not yet clear what the relation is between costs and benefits and how they can be transformed into a business model. In the social area, there is still a lack of research on the V2G technology, making it initially necessary to provide knowledge for disseminating the technology among possible users. This will be the main issue for future viability. Finally, environmental regulation is a major barrier to this technology. Several aspects have a direct impact on economic viability and the business model, such as the value of taxes and the lack of regulation regarding the use of electric energy storage technologies.

In this work, it was possible to divide and to characterize V2G in terms of both the demand side and the supply side, always based on the four main dimensions: technical, economic, environmental, social, and political. With this division, the technology was detailed in different subcategories, thus allowing for a more comprehensive view of the negative impacts, benefits, and barriers to energy sustainability improvements.

Considering the technical and economic scopes, the variety of factors analyzed corroborate improvements for both the demand and supply side. Points such as frequency regulation, peak shaving and seasonal energy storage are well covered by V2G on the supply side. On the demand side, awareness, and better knowledge on the part of users are necessary. Conversely, this technology is simple to operate and can offer the user a feeling of comfort about the availability of electrical energy—better quality energy and electrical supply in the event of a possible network outage.

In environmental terms, there are practically no impacts related to physical occupation, since the V2G is a small device installed inside the residential or commercial area. Regarding air emissions, the technology does not directly impact the environment, but indirectly, there can be a reduction in GHG emissions. The greatest terrestrial impact would be related to possible battery discharges. As the technology increases the charge and discharge ratio of the battery over time, greater degradation of these batteries can be generated, and this can

impact their disposal in the environment. It is worth mentioning that the impacts that this technology may have on battery life are not yet clear (discussed in item 5) [22,27].

Regarding the social dimension, further studies are still required to better understand how the technology can impact society and, thus, its advantages and disadvantages [7–10,16–22,27–29]. Social barriers to the penetration of V2G become an important aspect; problems such as high cost, connection to the power grid and problems with cybersecurity are points that require greater attention in future research.

Finally, regarding the political dimension, it can be considered that there is still little material covering regulation of the technology. The implementation of regulation in a market that contemplates the distribution, transmission and generation of electricity is a highly complex activity to be carried out in a short time. The creation of norms of responsibility for the use and interconnection of this technology with the electricity network becomes essential for the penetration of V2G in the electricity market. The regulations and standards that should be created to characterize the use of energy storage must be considered an integral part of the regulation and standardization for the use and introduction of V2G technology in society. Although the adoption of these new technologies is initially challenging, they bring evolution to society and to the country. Investments, job creation, specialized labor, environmental improvements, etc., are points that culminate in the development of the HDI, making it an important variable in the growth of social and governmental actions.

Having Santos as a pilot city, the insertion of this type of technology can be considered to have a long way to go. There is a high initial price to be paid due to the "technological novelty" issue and the necessary adaptations in all areas of society, but it will be extremely important and rewarding in the future of energy sustainability, considering technical, social, political, and environmental development.

Regarding electric vehicles, they are estimated to account for at least 40% of new worldwide sales by 2040, if this high percentage is reached mainly due to the quest (through environmental laws and regulations) for stabilizing the concentration of greenhouse gases at 450 ppm [30]. With the increase in the circulation of electric vehicles, the growing electricity demand is inevitable; therefore, technologies that can store and redistribute electricity will be inevitable for the future [31].

In conclusion, this paper addressed most of the impacts of V2G in different areas. Currently, environmental and technical–economic dimension data are less complicated to obtain and/or to estimate. Regarding the social and political dimensions, further research in these two areas is still needed, especially regarding laws, technical and market regulations, considering the role of consumers and energy companies in the negotiations for the purchase and sale of electricity inserted into the distribution network with this technology; there should also be an increase in the benefits and losses for society. However, this lack of accurate information in these areas is completely understandable and acceptable given the novelty of the V2G technology.

**Author Contributions:** M.F.B., P.R., M.E.M.U. and J.A.B.G. contributed to the whole article. All authors have read and agreed to the published version of the manuscript.

**Funding:** This research received no external funding.

**Institutional Review Board Statement:** Not applicable.

**Informed Consent Statement:** Not applicable.

**Data Availability Statement:** Not applicable.

**Conflicts of Interest:** The authors declare no conflict of interest.

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
