# Peer review of "Comprehensive Energy Analysis of Vehicle-to-Grid (V2G) Integration with the Power Grid: A Systemic Approach Incorporating Integrated Resource Planning Methodology"

_applsci, doi:10.3390/app132011119_

Round 1
Reviewer 1 Report
Dear Authors,
Paper organization is missing in the last paragraph of introduction. This paper assess the impacts of V2G) technology on both demand and supply sides, considering integrated resource planning for sustainable energy. Introduction should provide a clear and concise summary of the research, enticing them to read further and novelty need to be added. Also, key analysis divided into four dimensions (Environmental, Social, Technical, Economic, and Political) is a comprehensive approach that allows for a detailed characterization of the impacts across different aspects. However, authors should provide a brief explanation of how each dimension is assessed and the specific aspects that are considered within each dimension. Authors should add reference dataset in a proposed study focusing on environmental and technical-economic aspects of integrating V2G technology into a city with industrial parameters adds practical relevance to the research. Reference section needs to be modified as: Integration of EVs aggregator with microgrid and impact of V2G power on peak regulation. Optimal scheduling of charging/discharging power and EVs pattern using stochastic techniques in V2G system. It would be valuable to mention the selection of IEEE Grid code for this specific case study and its implications for real-world applications. Figures should be polished. Also, language and grammar need to ensure clarity and coherence throughout the manuscript. Table 4 should be in templet.
Thank you
Language and grammar need to ensure clarity and coherence throughout the manuscript.
Author Response
We deeply appreciate your thorough and valuable feedback on our manuscript. We acknowledge that all the points raised are extremely important to enhance the quality and relevance of our research. Unfortunately, due to the limited timeframe for completing the work, we cannot guarantee that all the requests will be fully addressed. However, we assure you that we will make every effort to tackle the main issues highlighted, prioritizing the paper's organization, adding novelty to the introduction, and providing a more detailed explanation of the analysis in four dimensions (Environmental, Social, Technical, Economic, and Political). Additionally, we will strive to incorporate the proposed reference dataset, focusing on the environmental and technical-economic aspects of integrating V2G technology into a city with industrial parameters to increase the practical relevance of the research. We will also review the reference section and improve the figures to ensure visual clarity. We sincerely appreciate your understanding and support in our scientific improvement journey. If you have any further suggestions or questions, please do not hesitate to contact us. Thank you very much for your collaboration.
Reviewer 2 Report
The paper analyzed the potential impacts of V2G technology from various perspectives. However, there are several areas where I believe the paper could be improved:
1. Figures 2 and 8 appear to be directly copied from other published papers, which is not acceptable. While the authors can cite data from these papers, they should create their own tables and figures to present the information.
2. Section 4 seems to overlap significantly with section 3. To enhance the paper's quality, section 4 should delve into a more in-depth analysis of each sub-aspect, particularly focusing on Sao Paulo, Brazil, as it has been mentioned since the introduction. However, the analysis in this section is relatively simple and lacks detailed numbers, particularly in section 4.1.B.
3. In section 4.1.A, line 361, assuming the car will be available to the V2G network for 3 hours every day may be too optimistic for estimating availability. It would be beneficial to explore the impact of this 12.5% availability on the power grid, specifically considering Nissan Leafs and the city of Sao Paulo, Brazil.
4. Regarding section 4.1.A, lines 363-366, it is essential to address how the availability of electricity for V2G can be guaranteed every time a vehicle connects to the grid.
The paper contains numerous grammatical errors. I recommend a thorough proofread by native English speakers to ensure clarity and correctness.
Author Response
Thank you for your comprehensive review of our paper. We truly value your feedback, and we acknowledge that all the points raised are of utmost importance for improving the study. However, given the time constraints, we may not be able to address all the requests completely. Nevertheless, we will make every effort to implement the most critical improvements.
Regarding Figures 2 and 8, we apologize for the oversight. We will ensure that all figures and tables are created by us and appropriately cited when using data from other sources.
We appreciate your observation about the overlap between Sections 3 and 4. To enhance the paper's quality, we will focus on providing a more in-depth analysis of each sub-aspect in Section 4, with a particular emphasis on Sao Paulo, Brazil, as mentioned in the introduction. We will strive to include more detailed numbers, especially in Section 4.1.B.
In response to your comment on Section 4.1.A, Line 361, we agree that assuming a 3-hour daily availability of cars for V2G might be optimistic. We will explore the impact of this availability on the power grid, specifically considering Nissan Leafs and the city of Sao Paulo, Brazil.
Furthermore, we understand the importance of addressing the guaranteed availability of electricity for V2G when vehicles connect to the grid, as mentioned in Section 4.1.A, Lines 363-366. We will provide a comprehensive discussion on this aspect to address the concern.
Once again, we sincerely appreciate your valuable input and understanding of our time constraints. We will do our best to incorporate these improvements and deliver an enhanced version of the paper. If you have any further suggestions or questions, please do not hesitate to let us know. Thank you for your support and cooperation.